# Application of Machine Learning Algorithms to Classify Peruvian Pisco Varieties Using an Electronic Nose

**DOI:** 10.3390/s23135864

**Published:** 2023-06-24

**Authors:** Celso De-La-Cruz, Jorge Trevejo-Pinedo, Fabiola Bravo, Karina Visurraga, Joseph Peña-Echevarría, Angela Pinedo, Freddy Rojas, María R. Sun-Kou

**Affiliations:** 1Department of Engineering, Pontifical Catholic University of Peru, Lima 15088, Peru; celso.delacruz@pucp.edu.pe (C.D.-L.-C.); fjrojas@pucp.edu.pe (F.R.); 2Department of Science, Pontifical Catholic University of Peru, Lima 15088, Peru; f.bravoh@pucp.edu.pe (F.B.); kvisurraga@pucp.edu.pe (K.V.); joseph.penae@pucp.pe (J.P.-E.); apinedo@pucp.edu.pe (A.P.); msun@pucp.edu.pe (M.R.S.-K.)

**Keywords:** artificial neural network, random forest, support vector machine, electronic nose, gas sensors array, beverage quality

## Abstract

Pisco is an alcoholic beverage obtained from grape juice distillation. Considered the flagship drink of Peru, it is produced following strict and specific quality standards. In this work, sensing results for volatile compounds in pisco, obtained with an electronic nose, were analyzed through the application of machine learning algorithms for the differentiation of pisco varieties. This differentiation aids in verifying beverage quality, considering the parameters established in its Designation of Origin”. For signal processing, neural networks, multiclass support vector machines and random forest machine learning algorithms were implemented in MATLAB. In addition, data augmentation was performed using a proposed procedure based on interpolation–extrapolation. All algorithms trained with augmented data showed an increase in performance and more reliable predictions compared to those trained with raw data. From the comparison of these results, it was found that the best performance was achieved with neural networks.

## 1. Introduction

The pandemic has caused a reinvention of commercial relationship dynamics, giving rise to developments in niche markets. For the alcoholic beverages industry, this is leading to the diversification of new products and a tendency towards selectivity in those already established [1]. In Peru, the outlook is equally encouraging, as this industrial sector has shown a growth of more than 30% since the pandemic, with sales of more than 4 million boxes of distillate. This has allowed it to recover the growth trend that it had previously. Additionally, the recovery of the international market has been supported by the migration of the sales segment towards higher-quality products [2]. This preference for premium products could be a strong opportunity to consolidate the pisco industry, but for this, it is necessary to develop a technology that facilitates its control and differentiation at a low cost.

Peruvian pisco is an alcoholic beverage with a unique flavor and aroma. This drink is made by distilling the fermented juice of certain varieties of pisco grapes, whose farming requires nutrients that come from soil in certain regions where they are grown, as well as by the type of distillation used in its preparation [3]. This distillate is relevant, since it is associated with our national identity; it is considered a Flagship Product and part of the Cultural Heritage of the Nation, and is recognized by the World Intellectual Property Organization as a designation of origin of Peru [4]. Therefore, pisco preparation requires the use of high-quality grape varieties (*Vitis vinifera*) in accordance with the denomination of origin. According to the aforementioned standard, pisco’s composition includes ethanol (range between 38 to 48%), which comes from the fermentation of glucose; methanol (levels up to 150 mg/100 mL of anhydrous alcohol), which comes from the hydrolysis of grape pectins, and is generally present in greater amounts in aromatic pisco compared to nonaromatic ones; and higher alcohols (propanol, isopropanol, butanol, iso-butanol, iso-pentanol and tert-pentanol), which also contribute to the aroma and flavor. However, high concentrations of these latter alcohols are unfavorable to the flavor. That is why, in the NTP 211.001 standard [5], a range of 60 to 350 mg/100 mL was established for higher alcohols. Most of these alcohols are produced from the reaction of amino acids by yeasts [6,7].

In addition, esters are a group of compounds that contribute floral and fruity aromas to pisco. These compounds are obtained from the reaction of acetyl-CoA (a coenzyme that is part of numerous metabolic pathways) with higher alcohols formed by the degradation of amino acids or carbohydrates [8]. Esters are also formed by condensation of acetic acid. Although the presence of the latter is important, it has a pungent and unpleasant odor, so an excess of this compound will cause an increase in the concentration of ethyl acetate, promoting an undesirable flavor in the distillate [9]. Likewise, Hidalgo et al. [10] observed that as fermentation levels increase, the concentration of acetic acid decreases, so this fermentation process is crucial. Aldehydes (such as acetaldehyde) are also produced during the fermentation process from ethanol by decarboxylation of pyruvic acid. This aldehyde also produces acetic acid by oxidation. It is important to control oxidative reactions because high aldehyde concentrations can cause off-flavors [11]. Another aldehyde present is furfural, produced during the distillation of residual sugars, especially when the heating stage is prolonged [7].

Currently, two methods are being used for the evaluation of the quality of pisco: organoleptic analysis through tasting and chemical analysis using instrumental analytical techniques, considering that a large percentage of pisco is composed of volatile organic compounds (VOCs). The tasting-based organoleptic analysis is performed by a personal expert known as a sommelier or pisco taster, who judges the quality and aroma of a pisco by taste and smell, while for the physicochemical analysis, the instrumental technique used is gas chromatography (GC) [5]. However, these techniques are not always viable, as there are differences between individuals performing organoleptic assays, and while instrumental techniques are accurate and reliable, they tend to be very expensive and time-consuming, and require being operated by an expert. It should be noted that other studies performed by gas chromatography (GC) systems achieved the complete characterization of the Quebranta and Italia varieties of pisco [12,13], as well as Moscatel and Torontel (aromatic varieties) [14] and Negra Criolla, Mollar and Uvina (nonaromatic varieties) [15].

In this project, we seek to apply augmentation and machine learning algorithms for the analysis of data obtained from an electronic nose with noncommercial sensors.

### 1.1. Gas Sensors

Electronic noses are devices made up of an array of gas sensors of different natures and compositions that allow for the identification of different volatile organic compounds (VOCs) found in complex gaseous samples. Considering the recent advances in nanotechnology, the design of sensors, circuits, and neural networks, their use has been proposed as an economical and portable technology to analyze the quality and possible adulterations in products produced in the food industry, as well as to carry out environmental monitoring. Among the many applications of electronic noses are the analysis of flours and grains, for example, in the detection of vapors originating from fungi in corn crops [16], in the diagnosis of microbial contamination in a variety of foods [17] and in determining the shelf life of nutritional cookies [18]. Because many VOCs are emitted during the ripening and rotting of fruits and vegetables, electronic noses have also been used to assess the freshness of fruits such as pears [19] and strawberries [20]. The oxidation of oils causes a degradation in their quality; thus, the oxidation and adulteration of olive oils can be determined using electronic noses [21], and even the geographical origin of olive oil can be identified based on its aromatic profile [22]. This equipment also finds application in the beverage industry, by monitoring quality, detecting adulterations, and identifying the geographic origins of coffee [23] and tea [24], as well as estimating beer quality in real time [25]. Like in the mentioned references, the electronic nose that we have developed aims to be used for distinguishing commercial pisco samples. In our case, we will take advantage of the different volatile compositions of pisco samples to differentiate them, so that the electronic nose can be employed in quality control. However, our electronic nose equipment differs from the aforementioned ones because it uses noncommercial sensors and employs a novel data-processing method in electronic noses.

The analysis of information collected by an electronic nose is usually performed by different multivariate statistical methods, such as principal component analysis (PCA) [26,27] and machine learning algorithms. They can predict the presence of certain compounds in the analyzed product, which may be an indicator of quality or adulteration. Among the most popular machine learning algorithms that provide good performance are artificial neural networks (ANNs) [28,29], multiclass support vector machines (MSVM) [30] and random forests (RF) [31]. To improve the training of these algorithms, a data augmentation procedure can be used.

Data augmentation is a common procedure in image processing [32,33] that generates synthetic data by scaling, rotating, panning, and warping the image. A more general way to increase the amount of data is through the feature space, for which new vectors are generated from the original feature vectors by adding noise, interpolating or extrapolating [34,35]. Rotation, translation and deformation algorithms make sense when working with images. However, when working with time series, it is convenient to work in the feature space, considering, for example, each element of the array as an element of the feature vector. Data augmentation in time series could be understood as obtaining new data from the temporary displacement, change in amplitude and deformation of the time series curve. Data augmentation in time series can also be performed using the synthetic minority oversampling technique (SMOTE) [36], which generates synthetic data by interpolating the original data. However, this technique has limitations in blinding samples. In order to improve this algorithm, Lyu et al. [37] proposed generating synthetic data by adding Gaussian noise and performing signal stretching. Augmented data can also be generated by performing a magnitude offset, magnitude warping or time warping, or combining both [38]. Increased data in the frequency domain can be generated for time series with a large content of frequency components such as electroencephalographic (EEG) signals [39].

Because pisco is a distilled beverage composed of a mixture of various VOCs, this research focused on the detection of specific VOCs contained in each variety of pisco, in order to determine the differences between the beverages in accordance with standards established in the “Designation of Origin”, regulated by INDECOPI (Peru’s national consumer protection authority) [4]. With this objective, an array of sensors (electronic nose) based on semiconductor metal oxides (MOS) was developed. These sensors were sensitive enough to detect and differentiate VOCs (mainly the majority ones) that are present in the aroma of each pisco variety [40]. The interaction of each sensor with the different VOCs of the sample generates signals that were recorded with a 1 Hz sampling frequency. The processing of the sensor responses, collected in a database, was carried out by applying several machine learning algorithms: ANNs, MSVM and RF. In this work, the analysis of the application of these algorithms was carried out with the objective of determining which of them provides the best performance.

This work yielded four meaningful contributions. First, we proposed a data augmentation algorithm that provides synthetic training data, with which the tested machine learning algorithms showed a general improvement in classification performance. Secondly, we observed a significant increase in the prediction accuracy for pisco varieties when an electronic nose with a greater variety of sensor types was considered. Our third contribution was demonstrating the superior performance of the RF and ANN algorithms over the MSVM algorithm for classifying pisco varieties. Fourth, and lastly, we determined the optimal collection of sensors for this task, in comparison with two other groups.

### 1.2. Machine Learning Algorithms

There are numerous machine learning algorithms, such as k-nearest neighbors (KNN), decision tree [41], the adaptive network based on fuzzy inference systems—ANFIS [42], MSVM [30], RF [31] and ANN [28,29,43], which are used in data analysis and modeling of chemical processes. The procedures with the best results in signal processing in electronic noses are detailed below.

In the investigation carried out by Hou et al. [31], the identification of Chinese liquors was carried out using an electronic nose designed for that purpose, and by processing the signals with various machine learning algorithms. The electronic nose used 10 metal oxide semiconductor (MOS)-based gas sensors, which were sensitive to alcohol and other liquor-related volatile organic compounds. Sensor signals were registered by an acquisition circuit for 360 s. A total of 8 types of Chinese liquors were used, with which 30 trials were performed for each liquor, with a period of 6 min per trial. In the work, they obtained the best results with the RF (93.8% accuracy), support vector machine—SVM (92.5% accuracy) and Bayesian—NB (90% accuracy) algorithms. In addition, the authors proposed an algorithm that they called double triangular feature-based sensor sequence (DTSS), with which they reached 96.3% accuracy.

The work carried out by Zhang et al. [30] proposed the use of an electronic nose for the classification of six Whiskey brands. Around 66 trials were conducted with each brand of whiskey. The authors evaluated five machine learning algorithms: linear discriminant analysis (LDA), SVM, k-nearest neighbor (KNN), bagged trees (BT), and subspace discriminant (SUBD). Of these five algorithms, the best performance was obtained by SVM, with 82.05% accuracy.

The work of Qui and Wang [28] applied an electronic nose for the prediction of food additives in fruit juices such as benzoic acid and chitosan. The electronic nose was composed of 10 sensors based on a metallic oxide semiconductor (MOS). For the analysis of the sensing data, it was divided into 5 classes of concentration level for the benzoic acid additive and 5 classes of concentration level for the chitosan additive; 24 trials were performed for each class. The processing of the signals coming from the electronic nose was carried out using the extreme learning machine (ELM) algorithm, which is a hidden single-layer feedforward ANN with the analytical determination of the output weights, unlike the backpropagation algorithm. This way of determining the output layer weights allows the algorithm to better generalize the responses. The ANN trained with the backpropagation algorithm does not generalize well when there is insufficient training data. Moreover, it uses the SVM, RF and partial least-squares regression (PLSR) algorithms. The algorithms that gave the best results were ELM, with a correlation coefficient (R2) of around 0.9123 and root-mean-square error (RMSE) of approximately 0.251; and RF, with an R2 of around 0.9268 and an RMSE of around 0.1017.

### 1.3. SMOTE

This algorithm was first presented by Chawla et al. in [35] and then applied in other works, as in [36]. The method is an interpolation algorithm applied over one of the original data and one of the nearest neighbors.

Let X_i_ be one sample of the original data and X_j_ be one sample of the *k*-nearest neighbors of X_i_. The synthetic data X_N_ are generated as follows.
X_N_ = X_i_ + α (X_j_ − X_i_),(1)
0 ≤ α ≤ 1(2)
where α is a uniform random number. The algorithm generates *M* synthetic data for the original sample X_i_. Thus, to generate *M* synthetic data, one sample (X_i_) is randomly chosen over the original dataset. If the algorithm is applied *T* times, the number of synthetic samples are *N = T M*.

### 1.4. Gaussian Noise and Signal Stretching

Lyu et al. [38] proposed a model independent data augmentation method based on adding Gaussian noise to the original data and stretching them.

The Gaussian noise vector is generated using the following equation:(3)$$\varepsilon =\left[\begin{array}{c} \varepsilon_{1}\\ \varepsilon_{2}\\ :\\ \varepsilon_{n} \end{array}\right];\;\varepsilon_{l}\sim G\left(\mu ,\sigma^{2}\right)\mathrm{for}\;l\;\mathrm{from}\;0\;\mathrm{to}\;n$$
where μ is the mean and σ is the standard deviation, and *n* is the number of elements for one sample. The mean is μ=0.

Let X_i_ be one sample of the original data. The synthetic data X_G_ are generated as follows:(4)$$X_{G}=X_{i}+\varepsilon$$
where signal stretching can be attained using the stretch factor α<1. Let m be a reduced number of first elements of the time series X_i_, which is calculated as follows:(5)$$m=\frac{n}{\left(1+\alpha \right)}$$

Then, a time series X_S_ of n elements is obtained, interpolating the *m* elements.

One sample X_i_ is randomly chosen over the original dataset to generate one sample of synthetic data X_G_ and one sample of synthetic data X_S_. Thus, if the algorithm is applied *T* times, the number of synthetic samples is *N* = 2*T*.

## 2. Materials and Methods

### 2.1. Electronic Nose for the Differentiation of Pisco Varieties

The electronic nose used (Figure 1) was developed at the Chemistry Section of the Pontifical Catholic University of Peru, and consists of an arrangement of gas sensors based on mixtures in different ratios of metal oxides, such as SnO_2_, TiO_2_ and MoO_3_. These oxides were synthesized by a hydrothermal method from water-soluble reagents and doped Pt and Pd in 0.05% to 0.1% proportions [40]. Then, doped oxides were deposited over alumina substrates with interdigitated platinum electrodes (Figure 2a,b) and placed in the sensing chamber (Figure 2c).

Volatile organic compounds (VOCs) follow oxidation reactions in the surface of the metal oxides, and as a result, there is a change in the electrical properties (resistance) of the sensors. The analog signal was transformed to voltage through the circuit shown in Figure 2d, and recorded with a sampling rate of 1 Hz by a National Instruments USB 6213 analog/digital converter (ADC). Then, the data were stored, processed and classified using various machine learning algorithms.

Each trial consisted of a sample and purge cycle. Ambient air was used as the carrier gas to purge the system. Pisco samples at ambient temperature and 1 atm were carried from a bubbler into the sensing chamber of the e-nose for 80 s followed by 400 s of air purge. The working temperature of the sensing chamber was controlled to 220 °C, with a preheating time of 100 min prior to the analysis for temperature stabilization.

### 2.2. Proposed Data Augmentation Interpolation–Extrapolation Method

Since a large number of trials are required to obtain reliable training results, data augmentation was used to generate synthetic data from the data obtained in sensing trials using the electronic nose. In the present work, the combination of the interpolating [34] and extrapolating [31] data augmentation methods is proposed. The interpolation–extrapolation can be obtained using the following equations:X_N_ = X_i_ + α (X_j_ − X_i_),(6)
C_1_ < α < C_2_, C_1_ ≤ 0, C_2_ ≥ 1,(7)
where X_i_ and X_j_ are two input vectors to the neural network (training data) of the same class; α is a random number; C_1_ and C_2_ are constant real values. If C_1_ = 0 and C_2_ = 1, there is only interpolation; if C_1_ = 1 and C_2_ > 1, or C_1_ < 0 and C_2_ = 0, there is only extrapolation. When C_1_ < 0 and C_2_ > 1, there will be either interpolation or extrapolation depending on the random value. Graphically, it can be represented in the feature space, as can be seen in Figure 3. In the time domain, it would look like Figure 4. In the latter case, it is observed that not only are amplitude variations obtained, but also variations in leading or lagging of the signals.

To generate synthetic data, one sample (X_i_) is randomly chosen over the original dataset, and another sample (X_j_) is randomly chosen over the *k*-nearest neighbor of X_i_. The k-nearest neighbor is used to permit the algorithm to generate data augmentation locally. The algorithm is applied *N* times to generate *N* synthetic data.

### 2.3. Machine Learning Training and Testing

The data acquired and registered from the sensors of the electronic nose were divided into training data and test data. Data augmentation was performed over the training data, while the test data were not modified. After realizing the data augmentation, the machine learning algorithms were trained. The evaluated algorithms were ANN, MSVM and RF.

Finally, the algorithms were evaluated using the test data. The accuracy performance metric was used for evaluation.

## 3. Results

The ANN, MSVM and RF machine learning algorithms were used with three datasets [44], each one obtained from trials employing a different group of sensors.

The trials were performed using the following six Peruvian pisco varieties, shown in Table 1.

### 3.1. Results with the First Dataset

The input for the machine learning algorithm is 240 sample data from the sensors shown in Table 2.

The oxides that were used in the preparation of the sensors included tin oxide (SnO_2_). Tin oxide was synthesized in the laboratory following the MK or MF procedure. In both procedures, tin chloride was dissolved in water, and a base was added dropwise until reaching a pH between 9–10. In the MF method, the procedure took place at room temperature, while in the MK method, the solution was slightly warmed (40–50 °C) on a heating plate.

At the end of these procedures, solid tin oxide was obtained. The development of these sensors is described in detail in a previous work [40], where several sensors were compared, of which the three sensors considered in this section showed the best sensing result according to the analysis of principal components (PCA).

Each sensor registered a rising voltage response when it reacted with pisco (Figure 5). The voltage values were recorded every second; in total, 80 voltage values of this rising response were taken in each trial. Binding the responses of the three sensors results in 240 samples, which were the input to the machine learning algorithm.

Data from 72 trials (12 trials per class) were used, of which 42 trials (7 trials per class) were used for training and data augmentation and 30 trials (5 trials per class) for testing. There was no opportunity for data leakage between the test data and the training data. The testing data were selected randomly from the 72 trials and was not affected by the data augmentation procedure. For comparison, these data are the same for all the predictions of the ANN, MSVM and RF algorithms.

#### 3.1.1. Artificial Neural Network—ANN

The neural network consists of three hidden layers, each one with 50 neuron units (Figure 6). The neural network outputs are six, one for each pisco class (sample numbers shown in Table 1). Each output gives the probability that the input data belong to the class associated with this output. For example, if the output associated with the Italy Q class is 90%, there is a high probability that the input data belong to this class. In this way, finding the highest probabilities of the outputs, the prediction of the class will be obtained.

Neural network training was performed using data from 42 trials. The results of the training without augmented data are shown in Figure 7. It learns with good accuracy (83.3%); however, the confusion matrix of “Validation” is 66.7% accurate. This evidences the lack of training data to properly classify the test data into the six classes. The more training data it has, the better the neural network will be able to generalize its learning.

Data augmentation was performed using the proposed methodology, generating α from Equations (6) and (7) with a uniform probability distribution in the interval (−2,3). To generate a synthetic data vector, one data vector was randomly chosen over the entire dataset and another data vector was randomly chosen over the *k* = 3 nearest neighbor.

Figure 8 shows the training confusion matrices using 2000 augmented data. Not only was there 100% accuracy with the training data, but also with the validation data. This is because the validation data were also augmented with synthetic data.

Figure 9 shows the accuracy results of the prediction using the test data. The test data did not use synthetic data. Several cases are shown with different amounts of augmented data in training. It can be observed that the accuracy improved with the increased data, up to 76.33% on average with 2000 augmented data, and there was less variability between one training and another.

#### 3.1.2. Multiclass Support Vector Machines—MSVM

The output of this algorithm is a value between 1 to 6, with each value representing a class (Table 1). Thus, the prediction will return the number of the class.

The training was carried out using the data from 42 trials plus the data augmented with the same methodology as in the training of the ANN. The results are shown in Figure 10. In this figure, ten trainings were not shown for each augmented datum, because the results were the same when using the MSVM. The accuracy improves with the augmented data. Moreover, the highest accuracy obtained was 63.33% with 100 augmented data.

#### 3.1.3. Random Forests—RF

The output of this algorithm is a value between 1 to 6, with each value representing a class (Table 1). Thus, the prediction will return the number of the class.

The training was carried out using the data from 42 trials plus the data augmented with the same methodology as in the training of the ANN. The results are shown in Figure 11. The accuracy improved with augmented data, although greater accuracy was not necessarily obtained with more augmented data. The highest accuracy obtained was 72.33% with 2000 augmented data.

### 3.2. Results with the Second Dataset

The second dataset was obtained using the sensors from the first dataset plus two sensors: SnO_2_-TiO_2_-1/1-MF and SnO_2_-TiO_2_-1/1-MK. These two added sensors provided a lower ability to differentiate alcohol samples [40]; however, when added to the three sensors used previously, they improved the classification results of pisco varieties. The sensors used to obtain the second dataset are presented in Table 3.

An example of the registration of the rising voltage response of this second group of sensors is shown in Figure 12.

The same classes of piscos were used as in the first dataset, in addition to the same number of trials for each class. In Figure 13, Figure 14 and Figure 15, the results of the signal processing are shown. An 85.00% accuracy was obtained using an ANN trained with 500 augmented data, 66.67% accuracy using an MSVM trained with 100 augmented data, and 81.67% accuracy using an RF trained with 0 augmented data.

### 3.3. Results with the Third Dataset

This third dataset was obtained from tests using three sensors developed by the following method: It was determined that the Pt-doped sensors improved the sensitivity of SnO_2_ and the composite (SnO_2_-TiO_2_) 4:1, due to the generation of more active sites (surface oxygen vacancies) compared to the other manufactured sensors. The methanol concentration in the pisco sample is apparently the factor that allows a better differentiation during sensing.

The order of the sensors in terms of the greatest response obtained in the sensing of the mixtures was (SnO_2_/TiO_2_) 1:4 > (SnO_2_/TiO_2_) 1:2 ≅ (SnO_2_/TiO_2_) 4:1 > 0.1% Pt (SnO_2_/TiO_2_)4:1≅ 0.05% Pt (SnO_2_/TiO_2_)4:1 > 0.05% −0.05% Pt-Pd (SnO_2_/TiO_2_) 4:1. This behavior was related to oxygen deficiency.

The sensors used in the electronic nose have a specific composition, as shown in Table 4.

An example of the registration of the rising voltage response of this third group of sensors is shown in Figure 16.

The same classes of piscos were used as in the first dataset, in addition to the same number of trials for each class. Figure 17, Figure 18 and Figure 19 show the results of the signal processing. As can be seen, these sensors provide better results in the classification of the quality of Peruvian pisco, reaching 98.67% accuracy when using an ANN with 2000 augmented data in the training. The other algorithms achieved 100% accuracy with the MSVM and 500 or more augmented data, and 98.33% accuracy with the RF and 100 augmented data.

Additionally, with the third dataset, the variation of parameters C_1_ and C_2_ was evaluated. The calculation of these limits was made using a CV parameter that varied from 0.02 to 4 as follows:C_1_ = 0.5 − CV,(8)
C_2_ = 0.5 + CV,(9)
with various values of the CV parameter generated. For each of these values, the parameters C_1_ and C_2_ were calculated, from which 500 augmented data were obtained; the training of the ANN algorithm was carried out and the accuracy was calculated with the test data. Figure 20 shows that when CV is around 2.5, the accuracy is 100%. In contrast, when the CV is around 0.5, the accuracy is around 95.45%. When CV = 0.5, C_1_ = 0 and C_2_ = 1 are obtained, and the augmented data generation procedure uses only interpolation. When CV > 0.5, we use the interpolation–extrapolation method. The results show that it is better to use the second method. This suggests that an increase in the variability of the training data can better separate the classes to be predicted.

### 3.4. Comparison with other Data Augmentation Methods

In this section, the proposed interpolation–extrapolation method is compared with the SMOTE and “Gaussian noise and signal stretching” data augmentation methods. For this comparison, the second dataset and the ANN algorithm were used. This dataset was used because compared with the first dataset, it gives better performance. The third dataset was not considered, because it gives 100% accuracy, which would not permit comparisons. The ANN gives better performance; for this reason, this algorithm was selected for the data augmentation method comparison.

The SMOTE parameter used in the experiments was k = 3, and the Gaussian noise and signal stretching parameters were sigma = 0.05, alfa = 0.1, while the proposed method parameters were k = 3 and c1 = −2, c2 = 3.

Table 5 and Table 6 show the test performance results using different data augmentation methods. Table 5 was obtained by applying the test data that were used in the previous section experiments, while Table 6 was obtained by applying alternative test data. In Table 5, it can be seen that the best results were obtained with the proposed interpolation–extrapolation method and signal stretching. These results suggest that both methods give similar performance. However, Table 6 shows that the proposed method is superior.

## 4. Discussion

In Table 7, the results obtained with augmented data for different algorithms and datasets are presented. The algorithms used are presented in the previous section. Additionally, principal component analysis—PCA was experimented, obtaining a reduced number of dimensions $n_{v}$ of the dataset. This algorithm was combined with MSVM to separate the classes and obtain accuracy performance; thus, it can be compared with the other algorithms. The resulting algorithm is PCA-$n_{v}$-MSVM. Moreover, different numbers of hidden layers $n_{l}$ and neuron units $n_{u}$ of ANN were used in the experiments; these algorithms are denoted by ANN-$n_{l}$-$n_{u}$. The results of the algorithms ANN-3-50, MSVM and RF presented in Table 7 are analyzed in the previous section.

Table 7 shows that with the third dataset, better performance was obtained than with the other ones using any of the algorithms. Within these results, we have the ANN-5-75, MSVM and PC-6-MSVM algorithms with 100% accuracy. The reduction to two dimensions of the dataset using the PCA algorithm and then applying MSVM does not give better result than using directly MSVM. However, in some cases, the use of three dimensions in PCA-3-MSVM gives better results than directly using MSVM. The use of more dimensions in PCA-$n_{v}$-MSVM does not imply better results.

In general, ANN-3-50 provides better performance considering the mean of the best results for each dataset compared to the performance of the other algorithm (see Table 8), with 86.67% accuracy on average. In the work by Zhang et al. [30], the MSVM algorithm achieved better results, and in the work of Hou et al. [31], the RF algorithm was successfully used. In the present work, it was found that ANN is superior to these algorithms when applied to the signals generated by the electronic nose used in the present work, and if there are enough training data (see Table 7). To attain enough training data, a good alternative is to use the proposed data augmentation method.

In Table 7, it can also be seen that in most cases, 500 augmented data are required in the training to obtain the best performance in the tests.

Figure 13, Figure 15 and Figure 19 show that in some cases, a higher amount of augmented data does not mean better results. Using 100 or 500 augmented data, the highest performances were attained, but for more augmented data, the performance was a little reduced. Additionally, in Table 9, the standard deviations of the results when applying ANN-3-50 are presented. It can be observed that this variability decreases with the augmented data; this can be also observed in Figure 9, Figure 13 and Figure 17. This result implies that the augmented data, in addition to improving the performance of the trained algorithm, provide greater reliability of the algorithm’s prediction when using ANN. For the case of applying RF, the variability does not present a trend of the change in variability when varying augmented data.

The data augmentation procedure with interpolation–extrapolation proposed in the present work generates better results in the generalization of the prediction compared to the data augmentation procedure only with interpolation. This is a clear advantage over the SMOTE procedure [36], which is a method that generates data by interpolating samples (see Table 5 and Table 6). Moreover, the proposed method is better than the Gaussian noise and signal stretching methods [37] when applied to the signals of this electronic nose, as can be seen in Table 5 and Table 6.

The previous work [40] analyzed five sensors; between these, three sensors showed the best sensing result according to the analysis of principal components (PCA). The present work analyzed the group of three sensors and the group of five sensors, and found that using more sensors increased the accuracy of the pisco variety prediction; even though, in this case, the two added sensors had less ability to differentiate alcohol samples. Furthermore, the present work experimented with a third group of sensors, which were obtained by performing variations in SnO_2_ and TiO_2_ proportions with respect to the sensors of the other groups. In Table 7, it can be observed that the third group is more sensitive to pisco varieties compared to the first and second group of sensors.

## 5. Conclusions

The proposed methodology of data augmentation with random interpolation–extrapolation in the feature space provides better results compared to the SMOTE, Gaussian noise and signal stretching algorithms. Moreover, it improves the training of ANN, MSVM and RF when a small dataset is available. In addition, it is found that the reliability of the prediction is high by decreasing the variability that exists between one training and another of ANN.

In general, the best performance was obtained with the training of an artificial neural network—ANN with 3 hidden layers and 50 units in each hidden layer using 500 augmented data, compared with the MSVM, RF and PCA plus MSVM algorithms.

The dataset obtained with the third group of sensors of the electronic nose provided better results using any of the three machine learning algorithms, with more than 97% of accuracy.

The increase in sensors corresponding to the second dataset improved the results in the classification of the Pisco variety compared to the first group of sensors.

## Figures and Tables

**Figure 1 sensors-23-05864-f001:**
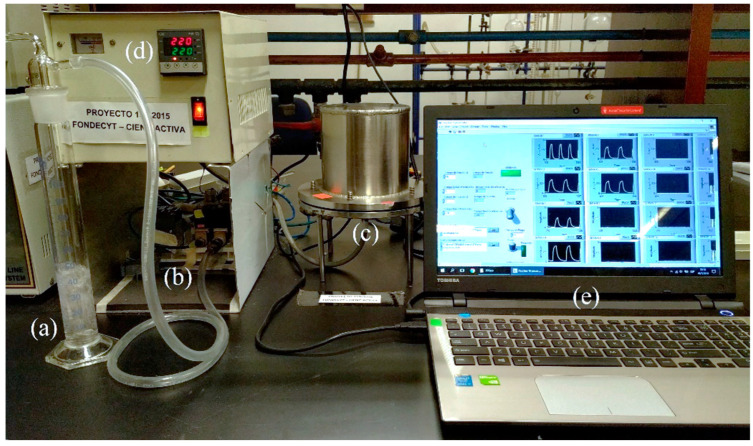
Electronic nose. (**a**) Pisco sample, (**b**) hydraulic system, (**c**) sensing chamber, (**d**) temperature controller, (**e**) LabVIEW software interface.

**Figure 2 sensors-23-05864-f002:**
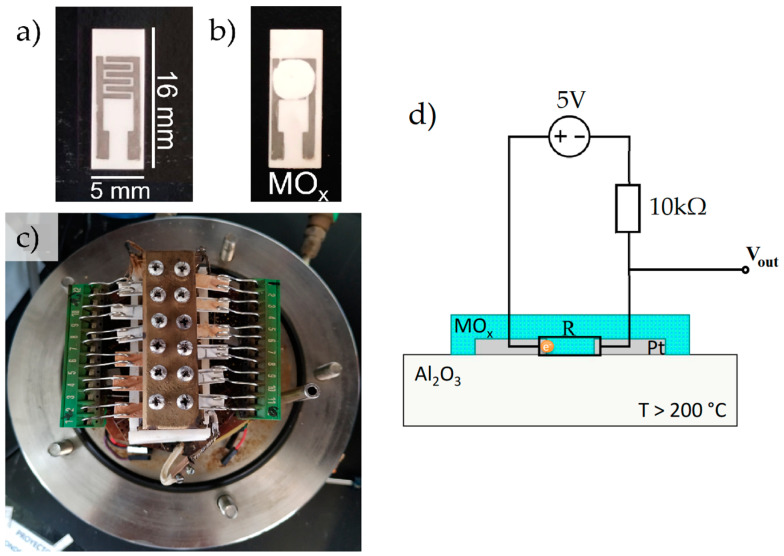
(**a**) Platinum electrodes over alumina substrate, (**b**) gas sensor prepared from a metal oxide (MO_x_), (**c**) arrangement of sensors inside the sensing chamber, (**d**) schematic representation of the sensor.

**Figure 3 sensors-23-05864-f003:**
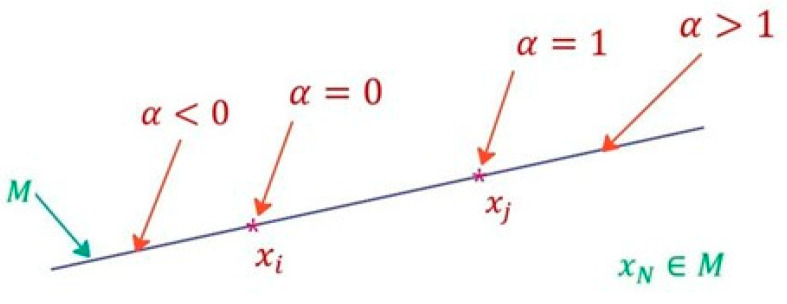
Interpolation–extrapolation in the feature space for data augmentation. M is the set formed by the points of the blue line.

**Figure 4 sensors-23-05864-f004:**
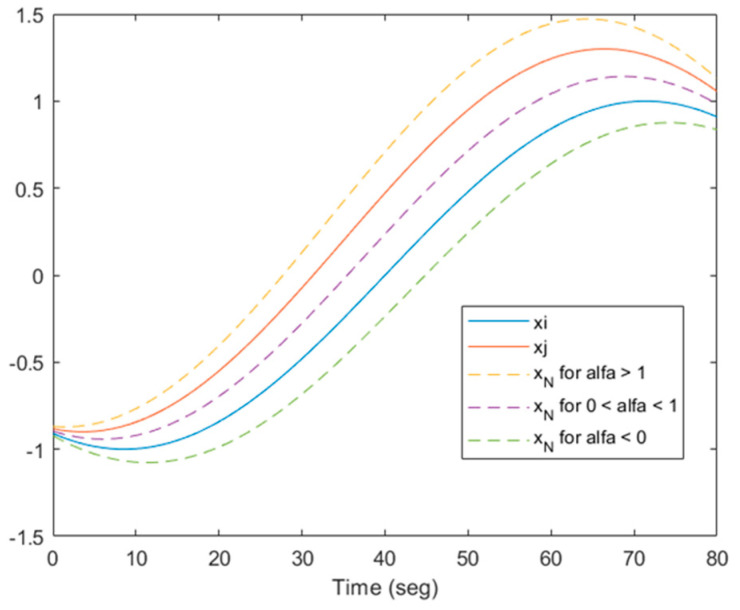
Interpolation–extrapolation in the time domain for data augmentation.

**Figure 5 sensors-23-05864-f005:**
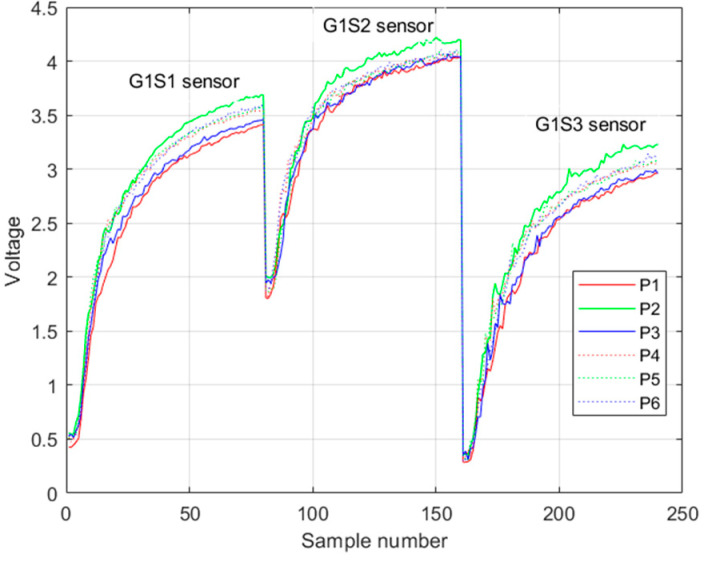
Example of the rising voltage response of the first group of sensors in one trial. Legend Pi is the i-th class of pisco variety.

**Figure 6 sensors-23-05864-f006:**
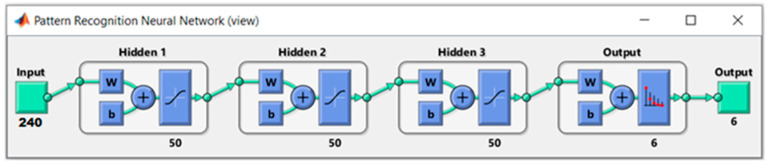
Structure of the neural network to classify into 6 classes (varieties and brands).

**Figure 7 sensors-23-05864-f007:**
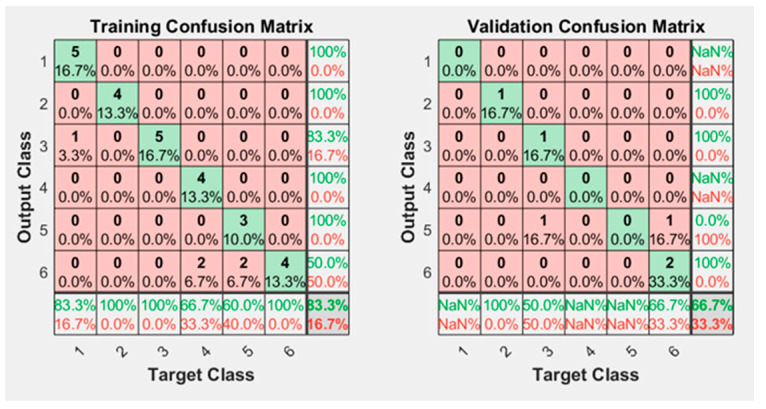
Confusion matrix obtained after training the neural network without using augmented data.

**Figure 8 sensors-23-05864-f008:**
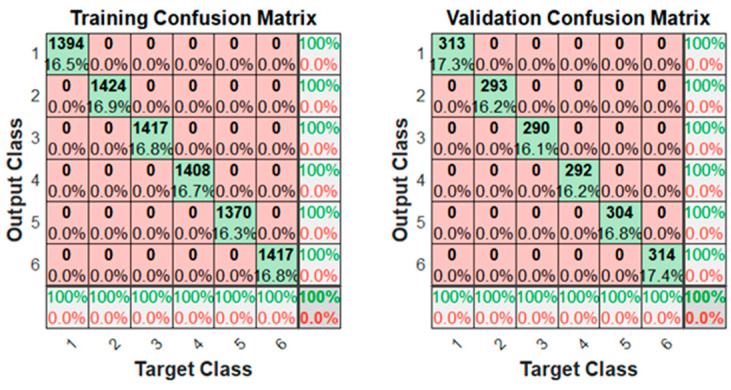
Confusion matrix obtained after training the neural network using 2000 augmented data.

**Figure 9 sensors-23-05864-f009:**
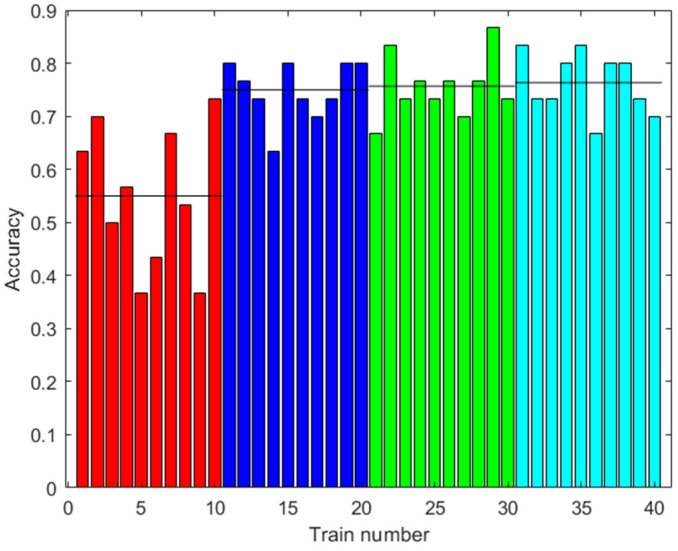
Results with the first dataset. Accuracy of the prediction of the ANN with the test data after training with different amounts of augmented data (0: red; 100: blue; 500: green; 2000: light blue). Ten trainings were generated for each case. The black line shows the mean accuracy for each case.

**Figure 10 sensors-23-05864-f010:**
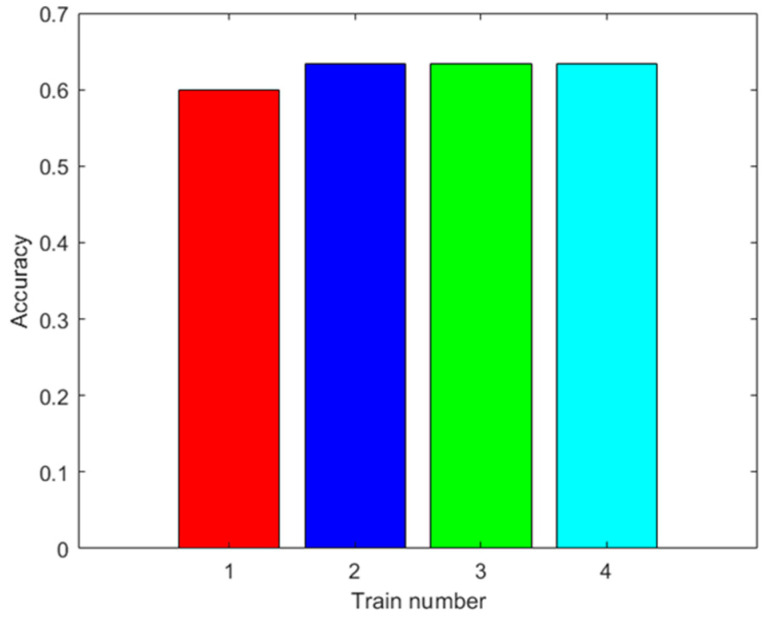
Results with the first dataset. Accuracy of the MSVM prediction with the test data after training with different amounts of augmented data (0: red; 100: blue; 500: green; 2000: light blue).

**Figure 11 sensors-23-05864-f011:**
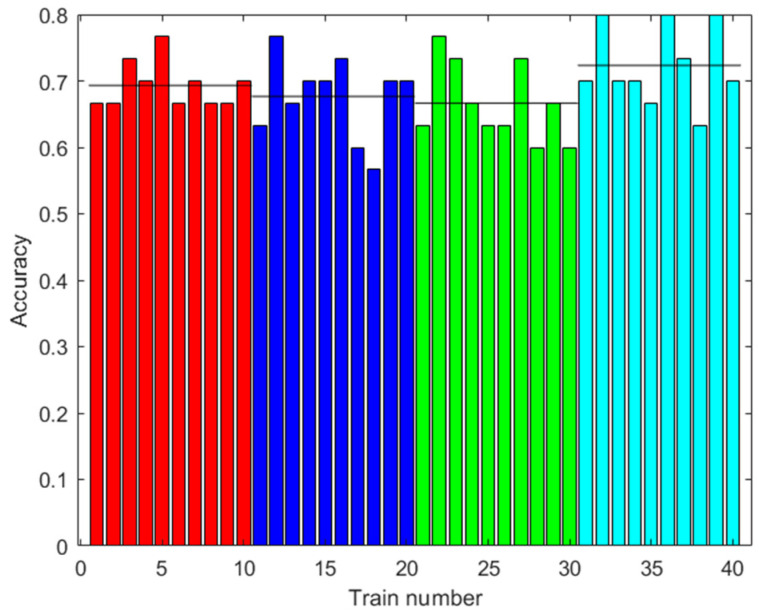
Results with the first dataset. Accuracy of the RF prediction with the test data after training with different amounts of augmented data (0: red; 100: blue; 500: green; 2000: light blue). Ten trainings were generated for each case. The black line shows the mean accuracy for each case.

**Figure 12 sensors-23-05864-f012:**
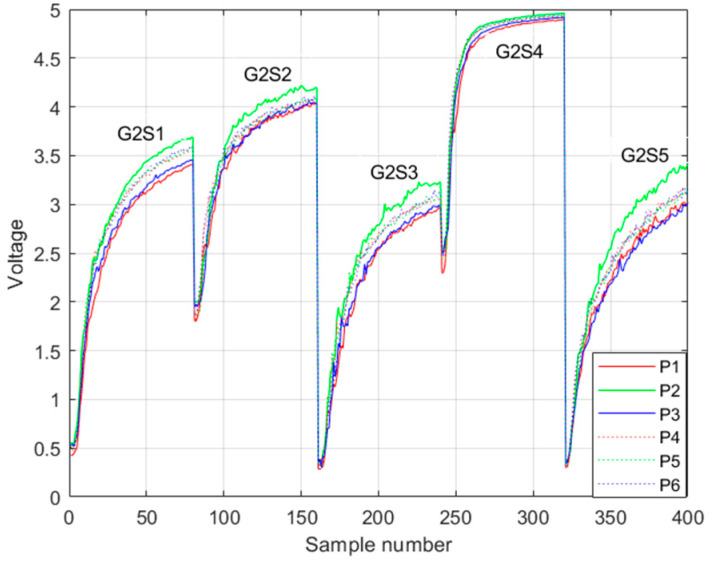
Example of the rising voltage response of the first group of sensors in one trial. Legend Pi is the i-th class of pisco variety.

**Figure 13 sensors-23-05864-f013:**
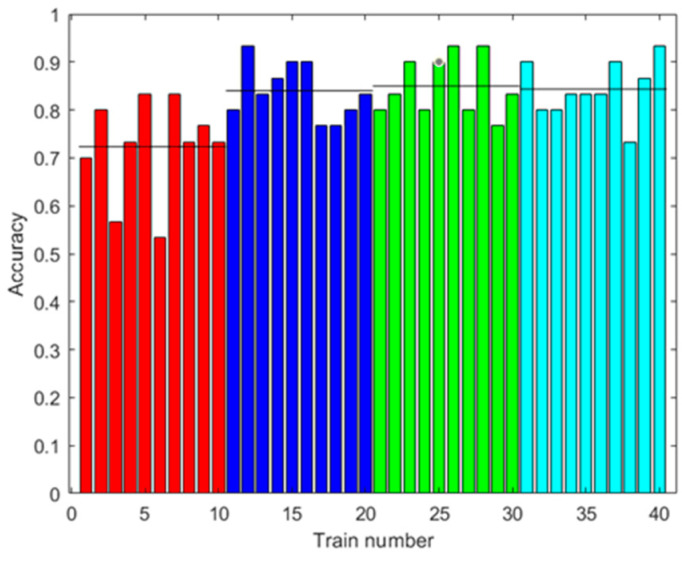
Results with the second dataset. Accuracy of the ANN prediction with the test data. Training with different amounts of augmented data (0: red; 100: blue; 500: green; 2000: light blue). Ten trainings were generated for each case. The black line shows the mean accuracy.

**Figure 14 sensors-23-05864-f014:**
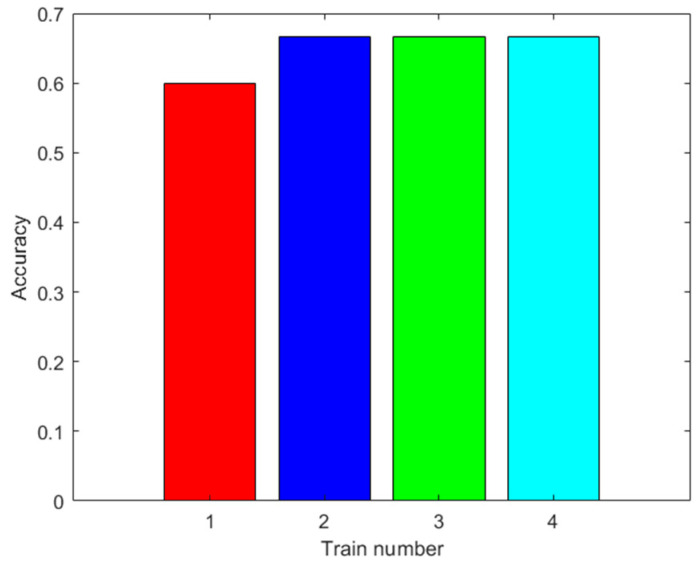
Results with the second dataset. Accuracy of the MSVM prediction with the test data after training with different amounts of augmented data (0: red; 100: blue; 500: green; 2000: light blue).

**Figure 15 sensors-23-05864-f015:**
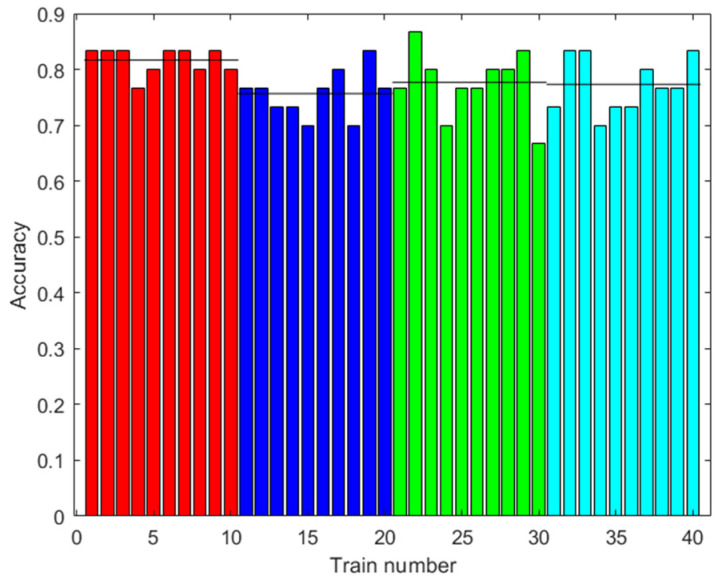
Results with the second dataset. Accuracy of the RF prediction with the test data. Training with different amounts of augmented data (0: red; 100: blue; 500: green; 2000: light blue). Ten trainings were generated for each case. The black line shows the mean accuracy.

**Figure 16 sensors-23-05864-f016:**
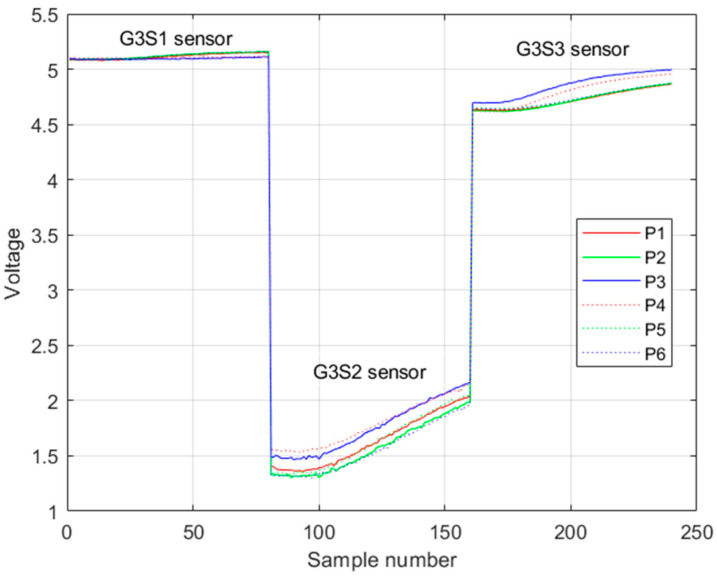
Example of the rising voltage response of the third group of sensors in one trial. Legend Pi is the i-th class of pisco variety.

**Figure 17 sensors-23-05864-f017:**
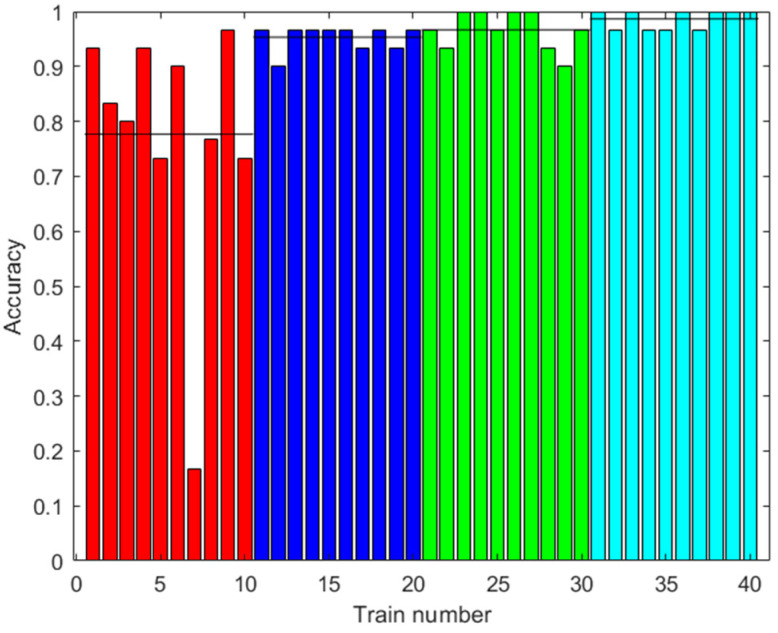
Results with the third dataset. Accuracy of the ANN prediction with the test data. Training with different amounts of augmented data (0: red; 100: blue; 500: green; 2000: light blue). Ten trainings were generated for each case. The black line shows the mean accuracy.

**Figure 18 sensors-23-05864-f018:**
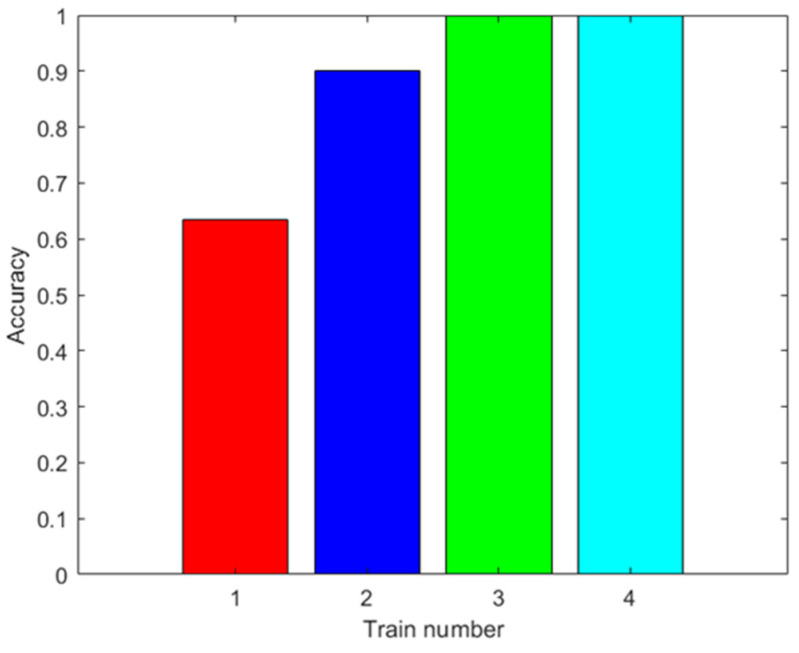
Results with the third dataset. Accuracy of the MSVM prediction with the test data after training with different amounts of augmented data (0: red; 100: blue; 500: green; 2000: light blue).

**Figure 19 sensors-23-05864-f019:**
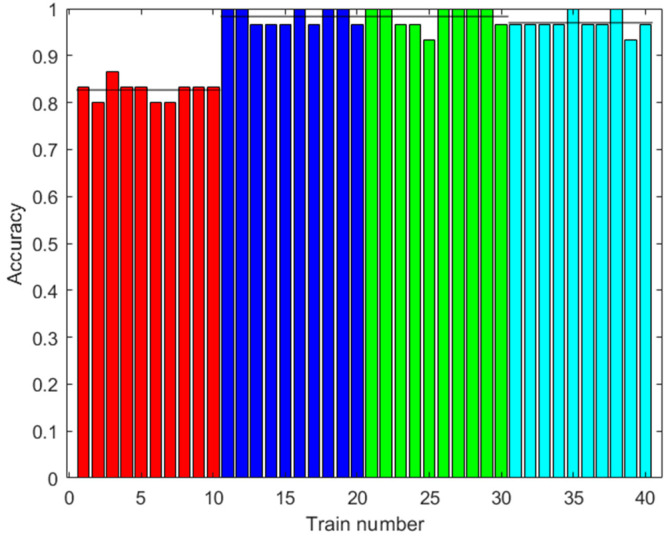
Results with the third dataset. Accuracy of the RF prediction with the test data. Training with different amounts of augmented data (0: red; 100: blue; 500: green; 2000: light blue). Ten trainings were generated for each case. The black line shows the mean accuracy.

**Figure 20 sensors-23-05864-f020:**
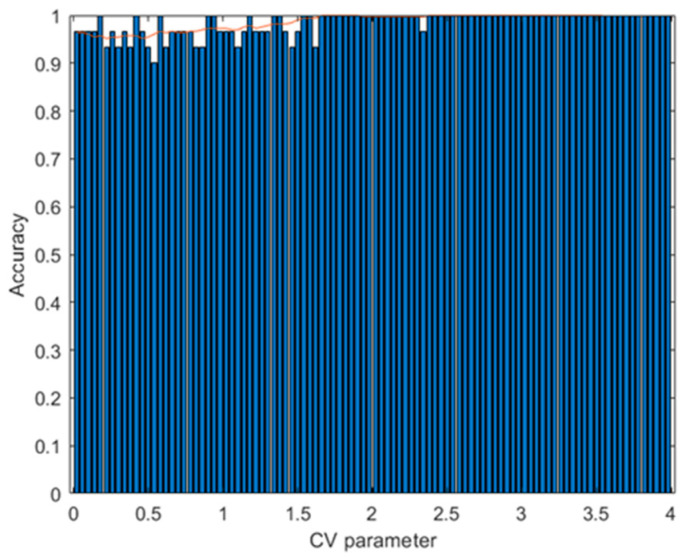
Results with the third dataset. Accuracy of the ANN prediction with the test data as a function of the variation of the CV parameter. Training with 500 augmented data for each CV value. The red curve indicates the average of the bar and 10 subsequent values.

**Table 1 sensors-23-05864-t001:** Peruvian pisco varieties analyzed.

Sample Number	Pisco Variety	Producer Abbreviation
1	Quebranta	T
2	Quebranta	D
3	Quebranta	Q
4	Italia	Q
5	Italia	T
6	Italia	D

**Table 2 sensors-23-05864-t002:** Sensors used in the electronic nose to obtain the first dataset.

Abbreviation	Composite Sensor
G1S1	(SnO_2_/TiO_2_)-1:2-MK
G1S2	(SnO_2_/MoO_3_)-1:1-MF
G1S3	(SnO_2_/TiO_2_)-1:2-MF

**Table 3 sensors-23-05864-t003:** Sensors used in the electronic nose to obtain the second dataset.

Abbreviation	Composite Sensor
G2S1	(SnO_2_/TiO_2_)-1:2-MK
G2S2	(SnO_2_/MoO_3_)-1:1-MF
G2S3	(SnO_2_/TiO_2_)-1:2-MF
G2S4	(SnO_2_/TiO_2_)-1:1-MF
G2S5	(SnO_2_/TiO_2_)-1:1-MK

**Table 4 sensors-23-05864-t004:** Sensors used in the electronic nose to obtain the third dataset.

Abbreviation	Composite Sensor
G3S1	(SnO_2_/TiO_2_)-4:1
G3S2	(SnO_2_/TiO_2_)-1:4
G3S3	(SnO_2_/TiO_2_)-1:2

**Table 5 sensors-23-05864-t005:** Results obtained with augmented data (AD) using the second dataset, the ANN machine learning algorithm and different data augmentation methods.

Data Augmentation Method	Accuracy (%)		
	100 AD	500 AD	2000 AD
Proposed	84.00	85.00	84.33
Signal stretching	84.67	86.67	85.00
Gaussian noise and signal stretching	80.67	82.00	84.00
SMOTE	74.67	77.00	78.00
Gaussian noise	76.33	73.67	70.67

**Table 6 sensors-23-05864-t006:** Results obtained with augmented data (AD) using the second dataset, the ANN machine learning algorithm, different data augmentation methods and alternative test data.

Data Augmentation Method	Accuracy (%)		
	100 AD	500 AD	2000 AD
Proposed	92.67	93.00	91.33
Signal stretching	84.33	84.00	87.67
Gaussian noise and signal stretching	83.33	83.67	86.67
SMOTE	88.33	88.67	85.67
Gaussian noise	84.00	87.33	85.67

**Table 7 sensors-23-05864-t007:** Results obtained with augmented data (AD) for different algorithms and datasets.

	1st Dataset—Accuracy (%)				2nd Dataset—Accuracy (%)				3rd Dataset—Accuracy (%)			
	0 AD	100 AD	500 AD	2000 AD	0 AD	100 AD	500 AD	2000 AD	0 AD	100 AD	500 AD	2000 AD
ANN-1-5	57.33	66.00	67.67	67.67	51.33	79.33	80.00	81.00	78.33	92.67	96.67	97.33
ANN-1-25	66.67	73.33	73.33	73.33	73.33	84.33	84.33	84.33	90.67	99.33	98.00	99.33
ANN-3-50	55.00	75.00	75.67	76.33	72.33	84.00	85.00	84.33	77.67	95.33	96.67	98.67
ANN-5-75	50.67	70.00	69.00	71.33	70.33	79.33	81.67	84.33	95.67	100.00	100.00	100.00
MSVM	60.00	63.33	63.33	63.33	60.00	66.67	66.67	66.67	63.33	90.00	100.00	100.00
RF	69.33	67.67	66.67	72.33	81.67	75.67	77.67	77.33	82.67	98.33	98.33	97.00
PCA-2-MSVM	56.67	43.33	46.67	50.00	43.33	40.00	50.00	46.67	56.67	73.33	80.00	80.00
PCA-3-MSVM	60.00	60.00	66.67	66.67	60.00	70.00	63.33	66.67	63.33	86.67	93.33	90.00
PCA-6-MSVM	60.00	53.33	60.00	43.33	53.33	50.00	53.33	53.33	63.33	90.00	100.00	100.00

**Table 8 sensors-23-05864-t008:** Mean of the best results for each dataset in Table 3.

	Mean
	Accuracy (%)
ANN-1-5	82.00
ANN-1-25	85.66
ANN-3-50	86.67
ANN-5-75	85.22
MSVM	76.67
RF	84.11
PCA-2-MSVM	62.22
PCA-3-MSVM	76.67
PCA-6-MSVM	71.11

**Table 9 sensors-23-05864-t009:** Standard deviation (SD) of the accuracy of different algorithms, datasets and augmented data (AD).

	1st Dataset—Accuracy SD (%)				2nd Dataset—Accuracy SD (%)				3rd Dataset—Accuracy SD (%)			
	0 AD	100 AD	500 AD	2000 AD	0 AD	100 AD	500 AD	2000 AD	0 AD	100 AD	500 AD	2000 AD
ANN-3-50	13.36	5.50	5.89	5.76	10.19	5.84	6.14	5.89	23.10	2.33	3.51	1.72
RF	3.44	6.10	5.88	5.89	2.36	4.17	5.89	4.92	2.11	1.76	2.36	1.89

## Data Availability

The data presented in this study are openly available in GitHub at https://doi.org/10.5281/zenodo.8072415.

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
