# Peer review of "Application of Machine Learning Algorithms to Classify Peruvian Pisco Varieties Using an Electronic Nose"

_sensors, 2023, doi:10.3390/s23135864_

Round 1
Reviewer 1 Report
This work is quite interesting. A few issues should be addressed:
1. There have been many recent articles in machine learning enabled E-nose for interesting applications, such as food identification, safety monitoring, environment gas identification and glucose detection. The authors should compare their work with these reported methods and analysis.
2. The effect of environmental factors, such as temperature and atmosphere, was not included. This factors will have very important impact on the precision.
3. The description of sample preparation and statistics of results should contain more details.
4. The authors should apply other methods, such as PCA, LDA and other machine learning techniques, to analyze the data. A detailed comparison among different methods will be helpful to judge he work.
5. Also, where is the dataset stored? Can the dataset in this paper be accessed by public and tested by other groups?
There is grammar issue.
Some long sentences can be separated into 2 - 3 shorter sentences.
Reviewer 2 Report
This paper presented a few algorithm combinations for pisco variety detections based on a sensor array arrangement, and their performance were analysed and compared. The application scenario for this work is important, but the general writing and results analysis need improving, thus I would only recommend it for publication after the following changes.
(1) Language and format: general writing needs tidy up; some long sentences can be separated for better understanding; some mixed up use of upper and lower cases; repeated sentences (e.g. line 322-324 and line 334-337); and also check grammar more carefully.
(2) In the experiment part, is the data recorded from the beginning of the experiment (pisco bubbling), or after the sensors are stabilized? Because from the sensor plots, it looks like the stabilization data were also used in the machine learning process, if so, the accuracy will be affected.
(3) The voltage response plot for the sensors (Fig 5 & 12), why are the three sensors plotted in sequential orders? Is this the way the data were recorded or just a format of plotting?
(4) Section 2.2 is a background literature review of some algorithms. This should be in the introduction. Also, the introduction can have subtitles, such as sensors, algorithms, etc, for the ease of reading.
(5) The ANN model described in this work has three hidden layers and 50 neurons each, why are these number of layers and neurons are chosen? Since for the number of the data this work is showing, it doesn’t necessary need this much calculation. This will only increase the computational power without increasing the accuracy.
(6) In Line 445-446, Zhang’s and Hou’s work were mentioned for comparison. But from what I understand, these work does not use the same sensor array, nor the same target substance (pisco in this case), nor the same dataset. So what’s the purpose of comparing the presented work with their work?
(7) SMOTE algorithm was mentioned in the discussion and conclusion to show the advances of the random interpolation-extrapolation algorithm. But SMOTE has not be introduced (one sentence in introduction is not enough), nor calculated, nor compared (if compared, not shown in this manuscript), so where is this conclusion comes from?
In general, the novelty of this paper is not very clear. The novelty can be the pre-process augmentation method, then the algorithms needs to be described clearer. Or if the novelty is this electronic nose system specialised for pisco detections, then the system as general needs to be shown. Also the comparison with other researchers’ work seems very random. The authors need to rearrange some sections, and rewrite some parts to tell a better story.
Same as Comment 1:
Language and format: general writing needs tidy up; some long sentences can be separated for better understanding; some mixed up use of upper and lower cases; repeated sentences (e.g. line 322-324 and line 334-337); and also check grammar more carefully.
Reviewer 3 Report
Sensors
Recommendation Acceptance: Minor Revision
Comment:
This manuscript introduced about application of machine learning algorithms to classify Peruvian Pisco varieties using an electronic nose. The trials were performed repeatedly for the accuracy of the analytical tool and the results were compared to previous research based on the analytical experimental tool. The authors suggested newly invented tool and the duplicated experimental results. However, there are some minor comments for better readability and provision of rigid and clear suggestion of the theme of this research. Though the manuscript was well organized with various graph and experiment data, there are some questions as follows:
1. Those machine learning algorithms utilized different amount of augmented data that is not adjustable for the comparison between various statistical methods.
2. There is only one architecture of Artificial Neural Networks model. Why don’t you suggest the structure of other models of MSBM and RF that were conducted through this research?
As a result, I recommend this manuscript for publication in Sensors after minor revision.

Round 2
Reviewer 2 Report
Thank you for addressing the comments and making the changes. I think this paper has now meet the standard for publication.